# How to Construct a Bottom-Up Nomothetic Network Model and Disclose Novel Nosological Classes by Integrating Risk Resilience and Adverse Outcome Pathways with the Phenome of Schizophrenia

**DOI:** 10.3390/brainsci10090645

**Published:** 2020-09-17

**Authors:** Michael Maes, Aristo Vojdani, Piotr Galecki, Buranee Kanchanatawan

**Affiliations:** 1Department of Psychiatry, Faculty of Medicine, King Chulalongkorn Memorial Hospital, Bangkok 10330, Thailand; drburanee@gmail.com; 2Department of Psychiatry, Medical University of Plovdiv, 4002 Plovdiv, Bulgaria; 3IMPACT Strategic Research Centre, Deakin University, PO Box 281, Geelong, VIC 3220, Australia; 4Immunosciences Lab., Inc., Los Angeles, CA 90035, USA; drari@msn.com; 5Cyrex Labs, LLC, Phoenix, AZ 85034, USA; 6Department of Preventive Medicine, Loma Linda University, Loma Linda, CA 92354, USA; 7Department of Adult Psychiatry, Medical University of Lodz, 91-229 Lodz, Poland; piotr.galecki@umed.lodz.pl

**Keywords:** deficit schizophrenia, cytokines, inflammation, neuro-immune, oxidative stress, leaky gut

## Abstract

Current case definitions of schizophrenia (DSM-5, ICD), made through a consensus among experts, are not cross-validated and lack construct reliability validity. The aim of this paper is to explain how to use bottom-up pattern recognition approaches to construct a reliable and replicable nomothetic network reflecting the direct effects of risk resilience (RR) factors, and direct and mediated effects of both RR and adverse outcome pathways (AOPs) on the schizophrenia phenome. This study was conducted using data from 40 healthy controls and 80 patients with schizophrenia. Using partial least squares (PLS) analysis, we found that 39.7% of the variance in the phenomenome (lowered self-reported quality of life) was explained by the unified effects of AOPs (IgA to tryptophan catabolites, LPS, and the paracellular pathway, cytokines, and oxidative stress biomarkers), the cognitome (memory and executive deficits), and symptomatome (negative symptoms, psychosis, hostility, excitation, mannerism, psychomotor retardation, formal thought disorders); 55.8% of the variance in the symptomatome was explained by a single trait extracted from AOPs and the cognitome; and 22.0% of the variance in the latter was explained by the RR (Q192R polymorphism and CMPAase activity, natural IgM, and IgM levels to zonulin). There were significant total effects (direct + mediated) of RR and AOPs on the symptomatome and the phenomenome. In the current study, we built a reliable nomothetic network that reflects the associations between RR, AOPs, and the phenome of schizophrenia and discovered new diagnostic subclasses of schizophrenia based on unified RR, AOPs, and phenome scores.

## 1. Introduction

Recently, we showed that the symptomatome of schizophrenia comprises psychosis, hostility, excitation, mannerism (PHEM), negative symptoms, formal thought disorders (FTD), and psychomotor retardation (PMR) and that these symptom domains are reflective manifestations of a single underlying trait, namely overall severity of schizophrenia (OSOS) [1,2]. Furthermore, we have shown that when OSOS increases, negative symptoms and PMR become more prominent and shape a distinct symptom profile, namely deficit schizophrenia [2]. The latter is delineated not only by negative symptoms and PMR, but also by PHEM symptoms [1,2,3,4,5].

Other key parts of the phenome of schizophrenia and deficit schizophrenia are neurocognitive impairments including executive functions, working memory, and semantic and episodic memory [2,3,5]. In addition, the impairments in these cognitive domains are reflective manifestations of an overall deficit in cognitive functioning, reflecting impairments in prefronto-temporal, prefronto-parietal, prefronto-striato-thalamic, and hippocampal and amygdalal neural circuits [1,3,5,6,7].

Furthermore, these impairments in the “cognitome” (the aggregate of cognitive dysfunctions) are strongly related with the symptomatome of schizophrenia and, based on the available knowledge, we concluded that these cognitome impairments may play a role in the development and maintenance of the symptomatome [1,5]. Lowered self-rated, health-related quality of life (HR-QoL) is another component of the phenome, namely the phenomenome, i.e., the description of the subjective experience from the patient [3,8]. Lowered HR-QoL in schizophrenia is strongly predicted by the symptomatome and impairments in the cognitome and is more pronounced in deficit schizophrenia than in non-deficit schizophrenia [3,8].

Increases in multiple immune and oxidative toxicities (MITOTOX) predict the impairments in the cognitome, symptomatome, and phenomenome [9]. These MITOTOX biomarkers include increased levels of neurotoxic tryptophan catabolites (TRYCATs), tumor necrosis factor (TNF)-α, a pro-inflammatory cytokine with neurotoxic effects; eotaxin (CCL11), an immune product with neurotoxic and anti-neurogenic effects; lipid hydroperoxides (LOOH) and malondialdehyde (MDA) indicating lipid peroxidation; advanced oxidation protein products (AOPPs), indicating increased chlorinative stress which exerts multiple toxic effects; increased IgM levels to zonulin, a toxic compound that leads to increased gut permeability; and increased bacterial translocation [1,9,10,11]. Furthermore, indicants of breakdown of the paracellular gut and blood brain barriers may aggravate these neurotoxic responses [10,11]. The combined effects of those toxic products of the immune-inflammatory response system (IRS) may directly (TRYCATs, TNF-α, AOPP, LPS) or indirectly (zonulin, breakdown of the paracellular barrier) lead to damaged neuronal functions in brain circuits, leading to impairments in the cognitome and the symptomatome [1,4,5,9,10,11].

Furthermore, some schizophrenia phenotypes are accompanied by deficits in the compensatory immune-regulatory system (CIRS), which upon activation may attenuate the IRS [12]. For example, patients with first episode psychosis (FEP) show a relative deficit in immune regulatory CIRS functions and increased IRS responses as compared to healthy controls [13]. Moreover, in deficit schizophrenia, additional impairments are detected in natural IgM-mediated autoimmune responses to oxidative specific epitopes (OSEs), which have strong anti-inflammatory, antioxidant, and antibacterial effects [9]. Finally, deficit schizophrenia is associated with the total paraoxonase 1 (PON1) status, namely a higher frequency of the Q allele and QQ genotype of the Q192R polymorphism in association with lowered activity of PON1 paraoxonase, an enzyme with anti-oxidative, anti-inflammatory, and antibacterial properties [14].

Causal reasoning indicates that the increased levels of zonulin as well as lowered IgM to natural OSEs and lowered PON1-gene associated paraoxonase activity may play a role in the breakdown of gut and blood brain barriers, bacterial translocation, immune activation, and oxidative stress, which together may induce neurocognitive toxicity and thus the phenome of schizophrenia [1,2,3,4,5,9,10,11,12,13,14]. Therefore, schizophrenia, and especially deficit schizophrenia, is accompanied by a disbalance in the causome (increased zonulin) versus the protectome (lowered natural IgM and PON1 paraoxonase activity) leading to lowered risk resilience (RR). Nevertheless, the direct effects of lowered RR on the adverse outcome pathways (AOPs, namely breakdown of the barriers and increased levels of neurotoxic products), the cognitome, symptomatome, and phenomenome of schizophrenia and the mediated (indirect) effects of RR and AOPs on the phenome have not been examined using the novel nomothetic network psychiatry (NNT) approach [9]. Furthermore, there is a lack of knowledge on how to construct nomothetic networks using Partial Least Squares (PLS) path modeling, which allows to estimate cause–effect path models, thereby combining principal component analysis to construct latent vectors coupled with multiple regression analysis.

Hence, this study aims to explain how (a) to construct a
reliable and replicable nomothetic network
of schizophrenia based on a theoretical model that was pre-specified based on our previous knowledge [1,2,3,4,5,9,10,11,12,13,14] (Figure 1) using feature sets extracted from RR, AOP, and phenome data; and (b) to discover a new classification of schizophrenia based on the same feature sets. Toward this end, we will explain how to use PLS path modeling as a bottom-up,
pattern recognition approach to construct a novel nomothetic network, which reflects the direct effects of the RR on the AOPs, and the direct and mediated effects of both RR and AOPs on the different features of the phenome. Subsequently, we will explain how latent variable scores may be computed from the feature sets and be used in unsupervised clustering techniques to disclose new classes built based on RR, AOP, and phenome features.


## 2. Subjects and Methods

### 2.1. Subjects

This study enrolled 80 schizophrenia patients and 40 healthy volunteers who were all Thai nationals of both genders and aged 18–65 years old. The schizophrenia patients were admitted to the outpatient OPD clinic of the Department of Psychiatry, Faculty of Medicine, Chulalongkorn University, Bangkok, Thailand. The diagnosis of schizophrenia was made using criteria of the Diagnostic and Statistical Manual of Mental Disorders (DSM), Fourth Edition Text Revision. They were all in a stable phase of illness and did not show psychotic flare-ups for at least one year prior to the study. Furthermore, we used the Schedule for Deficit Schizophrenia (SDS) [15] to make the diagnosis of deficit schizophrenia. We excluded patients with a current or lifetime diagnosis of other axis I disorders including bipolar disorder, a major depressive episode, autism spectrum disorders, generalized anxiety disorder, schizoaffective disorder, and substance use disorders (except tobacco use disorder). The healthy controls were family or friends of staff, or friends of patients, and they were recruited by word of mouth from the same catchment area in Bangkok, Thailand. We excluded controls with any current or lifetime axis-I diagnosis and controls with a positive family history of psychosis. Moreover, we excluded participants with (a) neurodegenerative and neuroinflammatory disorders such as Alzheimer’s and Parkinson’s disease, stroke, and multiple sclerosis; (b) immune and autoimmune disorders such as diabetes type I, chronic obstructive pulmonary disease, rheumatoid arthritis, psoriasis, and inflammatory bowel disease; (c) lifetime use of immunosuppressive and immunomodulatory drugs; (d) recent (6 months) use of antioxidants and ω3-polyunsaturated fatty acid supplements in therapeutic doses; and (e) pregnant and lactating women. All participants, as well as the guardians of patients (parents or other close family members) provided written informed consent to take part in the study. Approval for the study was obtained from the Institutional Review Board of the Faculty of Medicine, Chulalongkorn University, Bangkok, Thailand (No 298/57), which is in compliance with the International Guideline for Human Research protection as required by the Declaration of Helsinki, The Belmont Report, CIOMS Guideline and International Conference on Harmonization on Good Clinical Practice (ICH-GCP).

### 2.2. Clinical Assessments

We used the Mini-International Neuropsychiatric Interview (M.I.N.I.) in a validated Thai translation [16] to make the clinical diagnosis of schizophrenia. A semi-structured interview was used to collect clinical and socio-demographic data, such as psychiatric and medical history, age at onset, and duration of schizophrenia. We used different rating scales to score severity of negative symptoms, namely the SDS scale [15], the Scale for the Assessment of Negative Symptoms (SANS), [17] and the negative subscale of the Positive and Negative Syndrome Scale (PANSS) [18]. To score the PHEM domains, we assessed the items of the Hamilton Depression Rating Scale [19] and the Brief Psychiatric Rating Scale [20] and computed z unit-weighted composite scores as described previously [1,2,3,4,5]. Impairments in the cognitome were assessed with two instruments, namely the Cambridge Neuropsychological Test Automated Battery [21] and the Consortium to Establish a Registry for Alzheimer’s Disease (CERAD)-Neuropsychological [22] battery. Three key tests of CANTAB were used to compute an index of executive dysfunction [23]. In this study, we used three CERAD tests, namely the Verbal Fluency Test (VFT), Word List Memory (WLM), and Word List Recall, True Recall. The World Health Organization Quality of Life Instrument-Abbreviated version (WHO-QoL-BREF) [24] was used to assess HR-QoL. This scale contains 4 subdomains: (1) physical health; (2) psychological health; (3) social relationships; and (4) environment. Raw scores of the 4 domains were computed according to the WHO-QoL-BREF criteria [24]. We used DSM-IV-TR criteria to make the diagnosis of tobacco use disorder (TUD). Body mass index (BMI) was computed as body weight (kg)/length (m^2^).

### 2.3. Assays

The MITOTOX index was computed using the measurements of TNF-α, IL-6, CCL11, IgA to TRYCATs, IgA to LPS of 6 Gram-negative bacteria, lipid hydroperoxides (LOOH), malondialdehyde (MDA), and AOPP [9]. TNF-α, IL-6, and CCL11 were measured using the Bio-Plex^®®®^ 200 System (R&D Systems, Inc, Minneapolis, MN, USA) and IgA responses to TRYCATs and LPS were assayed using an enzyme-linked immunosorbent assay (ELISA) as described previously [9]. AOPP was assayed in a microplate reader (EnSpire, Perkin Elmer, Waltham, MA, USA) and LOOH was assayed by chemiluminescence in a Glomax Luminometer (TD 20/20) [9]. MDA was assayed using high-performance liquid chromatography (HPLC Alliance e2695, Waters’, Barueri, SP, Brazil) [9]. Breakdown of the (gut and BBB) paracellular barrier (IgA PARA) was measured and computed as described previously [10,11], namely the ratio between the paracellular pathway (assessed as a z unit-weighted composite score of occludin, claudin-5, E-cadherin, and β-catenin) and the transcellular pathway (assessed as a z unit-weighted composite score of talin, actin, vinculin, and epithelial intermediate filament [10]. Zonulin was purchased from Bio-Synthesis^®®®^ (Lewisville, TX, USA) and IgM to zonulin was assayed using an ELISA method [11]. IgM-mediated responses to the conjugated OSEs, MDA, azelaic acid, and Pi were analyzed as described previously and the optical densities were assayed at 492 nm using a multiscan spectrophotometer [9]. IgM levels were assayed using an immunoturbidimetric procedure with a kit purchased from Abbott (Chicago, IL, USA) using the Architect, model C8000, Abbott (Chicago, IL, USA). Total PON1 status, namely the PON1 Q192R genotype in an additive model and PON1 chloromethyl phenol acetate (CMPA)ase activity (PONgenozym) were analyzed using phenyl acetate (Sigma, St. Louis, MO, USA) under high salt condition and CMPA (Sigma, St. Louis, MO, USA) as substrates [14].

### 2.4. Statistical Analysis

We employed univariate GLM analysis to assess the differences in AOP, cognitome, symptomatome, and phenomenome data between subjects divided into those with a normal, lower, and very low RR. Post-hoc differences between these three groups were assessed using protected pair-wise comparisons among treatment means. In order to control for type 1 errors due to multiple comparisons, we used the false-discovery rate (FDR) procedure [25]. Associations between those groups and other nominal variables were assessed using contingency tables (χ^2^-tests) or Fisher’s exact test. Some (<0.05% of all data) causome, AOP or phenome data were missing, completely at random (MCAR), and these missing data were imputed using the series means method. All scale variables were standardized and the z-scores were used in the analyses. We have carried our neural network analysis using multilayer perceptron (MLP) models with diagnostic groups as output variables, an automated feedforward model with two hidden layers with up to 8 nodes and 250 epochs and mini-batch training with gradient descent and one consecutive step with no decrease in the error term as stopping criterion. Error, relative error, the area under the ROC curve, the confusion matrices, and the importance of the explanatory variables were computed, and the latter are shown in an importance chart. We used training (46.7%), testing (20%), and holdout (33.3%) samples. The above-mentioned tests were carried out using IBM SPSS 25 windows version.

We employed Partial Least Squares (PLS) path structural equation modeling (SmartPLS) [26] to delineate the most reliable nomothetic network explaining the paths from the causome ➔ AOPs ➔ cognitome ➔ symptomatome ➔ phenomenome. All these variables were entered as latent vectors (LV) extracted from their reflective manifestations (see below), except PONgenozyme, which was entered as a formative or reflective LV extracted from PON1 additive genetic model and PON1 CMPAase activity (thus reflecting PON1-gene-associated paraoxonase activity). Moreover, age, sex, BMI, and education were entered as single indicators. The phenomenome (HR Qol domains 1, 2, 3, and 4) was the final target which was predicted by all other indicators. We also examined possible moderator effects (interactions) among upstream LVs in predicting downstream LVs. We conducted complete PLS analysis on 5000 bootstrap samples only when the inner and outer models complied with specific quality data: (a) the model fit is adequate with SRMR <0.080; (b) all LVs have adequate composite reliability (>0.7), Cronbach’s alpha (>0.7), and rho_A (>0.8) and average variance extracted (AVE > 0.500); (c) all loadings on all LVs are >0.500 at *p* < 0.001; (d) the construct cross-validated redundancies and communalities are adequate as tested with Blindfolding; and (e) the discriminatory validity as checked with the Monotrait-Heterotrait index is adequate. Using 5000 bootstrap samples, we then computed the path coefficients with exact *p*-values and specific indirect, total indirect, and total direct effects. We used Confirmatory Tetrad analysis (CTA) to check whether the reflective model of the LVs is not mis-specified. Permutation and Multi-Group Analysis (MGA) were used to check whether there are any differences in the pathways between men and women. The latent variable scores obtained through PLS algorithms were used in subsequent analysis including clustering analysis.

We performed clustering analysis to classify the patients into relevant clusters based on the latent variable scores reflecting causome, AOPs, and phenome variables, and we employed the K-mean, K-median, and Forgy’s method using SPSS 25 and the Unscrambler (Camo, Oslo, Norway). These cluster-analytic procedures were used to disclose a new typology of stable phase schizophrenia based on all features of schizophrenia (bottom-up method). To further interpret the features of the cluster analysis-generated classes, we conducted analyses of variance (ANOVA), analysis of contingency tables (χ^2^-tests), and neural networks. The latent variable scores (all in z scores) were also displayed in clustered bar charts, with summaries of the separate variables in the cluster-generated groups.

## 3. Results

### 3.1. The Nomothetic Network Model to Be Tested

Based on our experience with pathway analysis and PLS path modeling [1,2,3,4,21], we will explain how to construct a nomothetic network using the causome (increased zonulin, the IgM protectome, and PONgenozyme), the AOPs (MITOTOX, IgA PARA, IgA TRYCATs, TNF-α), the cognitome (CANTAB executive test index, WLM, VFT, True Recall), the symptomatome (PHEM symptoms, SANS total score and PANSS negative subscale score) and the phenomenome (the 4 HR-QoL domains). We will show how to test whether the models created have construct validity.

The a priori assumption is shown in Figure 1. The Q allele coupled with its protein enzymatic activity (labeled as PONgenozyme), increased IgM to zonulin (labeled zonulin), and the natural IgM protectome (as assessed with IgM levels to azelaic acid, MDA, and PI, and total IgM, labeled natural IgM) may causally contribute to the phenome of schizophrenia. Increased zonulin is entered as a risk factor while PONgenozyme and natural IgM are part of the protectome. The phenome consists of different components, whereby the causome may induce a strain on the internal environmentome, which may develop into neurotoxic AOPs. The latter may induce neurocognitive toxicity in specific brain areas leading to deficits in the cognitome including in semantic and episodic memory, and executive functions. The combined effects of the causome, AOPs, impairments in the cognitome as well as socio-demographic data (sex, age, education) contribute to the clinical phenome of schizophrenia consisting of the symptomatome and the phenomenome.

### 3.2. Construction of a Risk Resilience Index Reflecting the Impact of Causome and Protectome

Based on the different components of the causome and protectome, we computed a z unit-weighted composite score reflecting the causome/protectome as: z (z IgM Pi + z azelaic acid + z MDA + IgM) + z (PC extracted from CMPAase activity + additive PON1 Q192R genetic model) − z IgM zonulin. Subsequently, we have split the population into three groups based on a visual binning method using z = −0.53 and z = 0.80 as cut-off-values. Table 1 shows the characteristics and demographic data of three groups, namely those with a low risk/high protection (normal risk resilience, RR), increased risk/lowered protection (lowered RR), and increased risk/very low protection (very low RR). There were no significant differences in age, sex, BMI, and TUD between the three groups. Education and employment rate were significantly lower in the very low RR group as compared with the other two RR groups. There were no significant associations between the RR groups and a familial history of psychosis. This table also lists the measurements of the biomarkers used to construct this RR index.

### 3.3. Lower Risk Resilience Predicts the AOPs

We found that the RR index was significantly correlated with MITOTOX (r = −0.355, *p* < 0.001, all n = 117), IgA PARA (r = −0.346, *p* < 0.001), CCL11 (r = −0.267, *p* = 0.003), IgA TRYCATs (r = −0.323, *p* < 0.001), and TNF-α (r = −0.249, *p* = 0.007). Table 2 shows the measurements of the biomarkers in the three RR groups indicating that MITOTOX, CCL11, and IgA PARA were significantly higher in the very low RR group compared with the other two groups. IgA TRYCATs were significantly higher in the very low RR groups as compared with the normal RR group. TNF-α was significantly higher in the low and very low RR groups as compared with the normal RR group. These group differences remained significant after FDR p-correction.

### 3.4. Lower Risk Resilience Predicts Impairments in the Cognitome

We found that the RR index was significantly correlated with WLM (r = 0.351, *p* < 0.001), VFT (r = 0.287, *p* = 0.002), True Recall (r = 0.316, *p* = 0.001), and executive functions (r = 0.292, *p* = 0.001). Table 2 shows the results of GLM analysis with the significant associations between the three RR groups and the cognitome measurements. The executive, WL Recall, and VFT scores were significantly lower in the very low RR group as compared with the other two groups. WLM was lower in the very low RR group as compared with the normal RR group. These group differences remained significant after FDR p-correction.

### 3.5. Lower Risk Resilience Predicts the Symptomatome

We found that the RR index was significantly and inversely correlated with psychosis (r = −0.360, *p* < 0.001), hostility (r = −0.316, *p* < 0.001), excitement (r = −0.426, *p* < 0.001), mannerism (r = −0.327, *p* < 0.001), FTD (r = −0.277, *p* = 0.003), PMR (r = −0.553, *p* < 0.001), PANSSnegative (r = −0.518, *p* < 0.001), and total SANS (r = −0.495, *p* < 0.001). Table 3 shows that all symptom domain scores (except hostility) were significantly higher in the very low RR exposome group as compared with the other 2 exposome groups. These group differences remained significant after FDR p-correction.

### 3.6. Lower Risk Resilience Predicts the Phenomenome (Lowered HR-QoL)

The RR index was significantly correlated with domain 1 (r = 0.276, *p* = 0.003), domain 2 (r = 0.249, *p* = 0.007), and domain 4 (r = 0.26, *p* = 0.005). Table 3 shows significant differences in domain 1, but not in domains 2, 3, and 4, between the three exposome groups. The domain 1 scores were significantly lower in the very low RR group as compared with the two other groups.

### 3.7. Construction of a First PLS Path Model.

Figure 2**.** Shows a first PLS-SEM pathway model which examined the causal paths from RR ➔ AOPs ➔ cognitome ➔ symptomatome ➔ phenomenome whereby each LV may predict one or more of the downstream LVs. In addition, age, sex, and education were also added to the model. RR was entered as three indicators, namely (a) zonulin as a single indicator, (b) a latent vector (LV) extracted from IgM to azelaic acid, MDA and Pi, and total IgM in a reflective model (labeled natural IgM protectome), and (c) a combination of PON1 CMPAase activity and the PON1 gene (additive model with QQ being 0 and RR being (2) in a formative or reflective model (labeled PONgenozyme). Nevertheless, in this PLS path model, zonulin was not significant and was thus deleted from the model. AOPs were entered as a reflective LV extracted from the 5 biomarkers scores. Nevertheless, CCL11 was not included as its loading was <0.5. The cognitome was entered as an LV extracted from the CANTAB/CERAD probe results (higher LV scores indicate more severe impairments) and the symptomatome was an LV extracted from all symptom domains used in this study. Finally, the phenomenome was entered as an LV extracted from the 4 HR-QoL domains. The overall fit of this PLS path model was adequate, with SRMR = 0.052. The construct reliability of the 4 reflective LVs was excellent with Cronbach’s alpha >0.755, composite reliability >0.845, rho_A > 0.788, and AVE > 0.580. All outer model loadings were >0.666 at *p* < 0.001 and the construct cross-validated redundancies and communalities were adequate. In addition, the discriminant validity was also adequate. Complete PLS pathway analysis with 5000 bootstraps showed that 40.9% of the variance in the phenomenome was explained by the regression on the symptomatome and the cognitome. The other paths from RR and AOPs to the phenomenome were non-significant. We found that 59.9% of the variance in the symptomatome was explained by the cognitome and AOPs, while 52.5% of the variance in the cognitome was explained by AOPs and 24.2% of the variance in the latter was explained by natural IgM and PONgenozyme. In this figure, we show a formative gene-enzyme activity model, however, a reflective model showed a similar result with loadings on both variables > 0.7.

There were significant specific indirect effects of PONgenozyme on (a) the cognitome mediated via AOPs (t = −2.41, *p* = 0.016); (b) the phenomenome mediated by the path from AOPs ➔ cognitome ➔ symptomatome (t = −2.21, *p* = 0.027); (c) symptomatome mediated via AOPs (t = −2.08, *p* = 0.037) and the path from AOPs ➔ cognitome (t = 2.36, *p* = 0.018). There were significant total effects of the PONgenozyme on (a) AOPs (t = −2.56, *p* = 0.010), cognitome (t = −2.41, *p* = 0.016), symptomatome (t = −2.41, *p* = 0.016), and the phenomenome (t = 2.46, *p* = 0.014).

There were significant specific indirect effects of the natural IgM on (a) the cognitome mediated via AOPs (t = −4.09, *p* < 0.001); (b) the symptomatome mediated by AOPs (t = 3.89, *p* < 0.001) and mediated by the path from AOPs ➔ cognitome (t = −3.37, *p* = 0.001); (c) the phenomenome mediated by the path from AOPs ➔ cognitome (t = 2.17, *p* = 0030), AOPs ➔ symptomatome (t = 2.82, *p* = 0.005), and AOPs ➔ cognitome ➔ symptomatome (t = 2.90, *p* = 0.004). There were significant total effects of the natural IgM on AOPs (t = 5.23, *p* < 0.001), cognitome (t = 4.09, *p* < 0.001), symptomatome (t = 4.70, *p* < 0.001), and phenomenome (t = 4.66, *p* < 0.001).

There were significant specific indirect effects of AOPs on (a) the symptomatome mediated by the cognitome (t = 5.36, *p* < 0.001); and (b) phenomenome mediated by the cognitome (t = 2.54, *p* = 0.011), the symptomatome (t = 3.13, *p* = 0.002), and the path from cognitome ➔ symptomatome (t = 3.77, *p* < 0.001). There were significant total effects of AOPs on the cognitome (t = 8.65, *p* < 0.001), symptomatome (t = 10.78, *p* < 0.001), and the phenomenome (t = 9.12, *p* < 0.001). There was also a significant total effect of the cognitome on the phenomenome (t = 6.94, *p* < 0.001). Education had a significant direct effect on impairments in the cognitome (t = 7.81, *p* < 0.001), symptomatome (t = −4.99, *p* < 0.001), and the phenomenome (t = 5.21, *p* = <0.001), while age had a significant total effect on the cognitome (t = 2.21, *p* = 0.027).

### 3.8. Construction of a Second PLS Path Model

In the second PLS path analysis (Figure 3), we have combined the AOPs and the cognitome into one construct extracted from 9 indicators (4 cognitive and the 5 AOPs) of an underlying single trait (the AOP-cognitome). In this model, zonulin was significant and thus included in the final model. The overall fit of this model was adequate with SRMR = 0.055 and the construct reliability of the AOP-cognitome LV was good with Cronbach’s alpha = 0.853, composite reliability = 0.886, rho A = 0.869, and AVE = 0.496. All loadings on this LV were >0.500 at *p* < 0.001. Blindfolding showed that the construct cross-validated redundancy (0.291) was sufficient and also the discriminant validity was adequate as ascertained with the Heterotrait-Monotrait ratio. Complete PLS path analysis with 5000 bootstraps showed that 39.7% of the variance in the HR-QoL phenomenome could be explained by the cumulative effects of the AOP-cognitome and symptomatome; 55.8% of the variance in the latter was explained by the AOP-cognitome and 22.0% of the variance in the latter by the cumulative effects of the RR components. There were significant indirect effects of (a) zonulin (t = 2.31, *p* = 0.021), PONgenozyme (t = 2.39, *p* = 0.017) and natural IgM (t = 4.91, *p* < 0.001) on the symptomatome, all mediated by AOPs; (b) zonulin on the phenomenome mediated by the path from AOP ➔ symptomatome (t = 1.99, *p* = 0.047); (c) PONgenozyme on the phenomenome mediated by the path from AOP-cognitome ➔ symptomatome (t = 2.08, *p* = 0.038); and (d) natural IgM on the phenomenome mediated by the AOP-cognitome (t = 2.13, *p* = 0.033) and the path from the AOP-cognitome ➔ symptomatome (t = 3.33, *p* = 0.001). There were significant total effects of the three RR factors on the phenomenome (all *p* < 0.05) and on the symptomatome (all *p* < 0.05). In this model, we have also examined whether there are sex-related differences in this nomothetic network using MGA and permutations. We found that there were significant differences in the total effects (*p* = 0.002) and the path from AOP-cognitome ➔ phenomenome with a significant impact in women (path coefficient = −0.475, *p* < 0001), but not in men (path coefficient = 0.021, non-significant).

### 3.9. Computation of Latent Variable Severity Scores

To compute scores reflecting the severity of RR, AOPs, cognitome, symptomatome, and phenomenome, we have conducted PLS analyses and calculated latent variable scores. In order to obtain an objective OSOS index, we computed a latent variable score of an LV extracted from all AOPs, cognitive test results, and all symptom domains. The latter LV showed adequate reliability validity with Cronbach alpha = 0.942, composite reliability of 0.951, rho_A = 0.954, and AVE = 0.569. Moreover, blindfolding showed that the construct cross-validated redundancy of this LV was adequate (0.231).

### 3.10. A New Bottom-Up Classification of Schizophrenia

Based on the different latent scores reflecting the severity of RR, AOPs, cognitome, symptomatome, OSOS, and phenomenome, we have performed different cluster analyses (K means, K-median, Forgy’s method), which yielded comparable results with a solution whereby patients are divided into two groups, a first cluster with 46 patients, and a second with 34 patients. There is a strong association between this new classification and the classification into deficit and non-deficit schizophrenia (χ^2^ = 40.1, df = 1, *p* < 0.001), although 9 patients with deficit schizophrenia were allocated to cluster 1 and 3 with nondeficit schizophrenia to cluster 2.

Figure 4. Shows a bar graph with the latent scores in healthy controls and the 2 clusters of schizophrenia patients. We analyzed the differences in latent scores between the three groups using univariate GLM analyses with age, sex, education, and BMI as covariates. Cluster 1 patients are differentiated from healthy controls by increased zonulin, AOPs, cognitome, symptomatome, and phenomenome scores (all *p* < 0.001). Cluster 2 patients are significantly differentiated from healthy controls by increased zonulin, lowered PONgenozyme and natural IgM, and increased cognitome, symptomatome, and phenomenome scores. Cluster 2 patients are also differentiated from cluster 1 patients by lowered PONgenozyme and natural IgM, and increased AOP, cognitome, symptomatome, and phenomenome scores. Univariate GLM analysis with sex, age, education, and BMI as covariates showed that OSOS is significantly different (F = 132.16, df = 2/108, *p* < 0.001; partial-eta squared = 0.710) between the three groups and increases from controls ➔ cluster 1 ➔ cluster 2 (all different at *p* < 0.001).

A neural network analysis with controls and both cluster analysis-generated groups as output variables and the RR, AOP, phenome, OSOS, and phenomenome scores as input variables showed an adequate discrimination between the three groups. A feedforward network with 8 input units, 2 hidden layers with 4 units in layer 1, and 3 units in layer 2 was conducted with 250 epochs and the activation function in the hidden layer was a hyperbolic tangent and in the output layer identity. The network information showed that the error term (sum of squares) and the percentage of incorrect classifications were lower in the testing (3.453 and 12.9%) than in the training (6.999 and 16.1%) sample, indicating that the model learned to generalize from the trend and is not over-trained. The AUC ROC was 0.947 for normal controls, 0.926 for cluster 2, and 0.999 for cluster 3. The confusion matrix showed an accuracy of 90.9% in the holdout sample. Figure 5 shows the importance chart and that the symptomatome, OSOS and AOP had the highest predictive power of the model, followed at a distance by PONgenozyme and the phenomenome, and again followed at distance by the cognitome, natural IgM, and zonulin.

## 4. Discussion

In the present paper, we explained how to construct a reliable and replicable nomothetic network that unifies the different building blocks of a major mental illness, namely a disbalance between risk and protective factors, the AOPs, cognitive impairments, clinical phenome, and phenomenology (namely lowered self-reported HR-QoL). Furthermore, this bottom-up model of schizophrenia was built based on inductive and causal reasoning through identification of the RR, AOPs, and phenome and ensembling data drawn from those observations into a unified causal model. This contrasts with the current gold standard use of case definitions of schizophrenia proposed by the APA (DSM-5) and the WHO (ICD). The latter diagnostic classifications are based on consensus criteria, which consider descriptive aspects of the disorder and narratives derived from observer-based interviews or self-reports by the patient [27]. Moreover, the DSM and ICD taxonomies have insufficient reliability and validity as indicated by differences in diagnoses using the DSM-III-R, DSM-IV, and ICD-8, ICD-9, ICD-10 classifications [27,28]. Using these diagnostic classifications, schizophrenia may be under-diagnosed or over-diagnosed [28] and there is variability in clinical diagnosis with inter-departmental diagnostic differences when using the ICD-8 and ICD-10 taxonomies [29]. This insufficient unification and harmonization of DSM and ICD diagnoses did not improve in recent DSM and ICD versions [27]. Therefore, it is suggested that these taxonomies lack validity and may even be counterproductive for research purposes [27,30,31,32]. Importantly, using machine learning techniques, we observed that schizophrenia comprises qualitatively distinct symptomatic entities, which were externally validated by biomarkers [3]. This underscores the poor description of clinical and biological heterogeneity of schizophrenia by the DSM [33]. Last but not least, the DSM and ICD taxonomies are not based on domain knowledge of a theoretical model underpinning the illness, which would allow a deductive, top-down-driven approach.

These non-validated taxonomies are then employed in top-down experiments [27] whereby the diagnostic groups (schizophrenia versus controls) are entered as input variables in t-tests, ANOVAs, or GLM analyses to detect changes in biomarkers, which are entered as output or dependent variables. Nevertheless, causal and inductive reasoning indicates that these RR and AOPs may explain the onset of cognitive disorders and psychosis and thus that those biomarkers should be used as input variables in, for example, logistic regression analysis, support vector machine, or neural networks predicting the target diagnosis. Because the DSM and ICD case definitions of schizophrenia are not reliable and do not account for the existing clinical and biological heterogeneity, it is not surprising that decades of biomarker research did not provide external validating biomarker criteria.

To overcome these problems, we have used a bottom-up, data-driven approach to examine how the RR indices may affect the AOPs and the phenome and additionally we assembled and integrated all these feature sets of schizophrenia to reify the diagnosis into a novel explicit data model. As such, the RR and AOPs are integrated and incorporated into a new, unified cause-to-outcome model of schizophrenia. This approach not only offers a more comprehensive picture of a disorder, but also objectivates the phenome of schizophrenia, thereby translating the RR and AOP features sets and cognitive features to psychiatric scores, and vice versa [27]. Our new explanatory modeling approach constructing nomothetic networks, therefore, not only allows to create new models using computer science (a method called reification), but also represents the learned information in a model that now treats a descriptive concept (the DSM and ICD diagnoses) as a material concept (this is also reification).

Moreover, it is also important to stress that the input RR and AOP features were re-engineered and pre-processed, including scaling and normalizing and that the standardized RR data were employed to compute z unit-weighted composite scores, which reflect the interactions between the causome and protectome, yielding a new risk resilience (RR) score. Likewise, we computed a z unit-based composite score on eight different biomarker systems which have shared neurotoxic activity (MITOTOX). As such, the latter score reflects the combined and interactive effects of multiple immune and oxidative stress pathways, which are known to cause neuronal dysfunctions including in neuroplasticity, neurogenesis, and cell death and apoptotic pathways [9]. Furthermore, this MITOTOX index was then combined with impairments in the paracellular gut and blood brain barriers, yielding a latent variable score that reflects their interactions. As such, we have reduced a larger number of biomarkers (n = 25) into one LV, which reflects the interactions among all pathways into one meaningful concept, namely AOPs. Moreover, our findings that one latent trait may be extracted from these biomarkers indicate that the latter are manifestations of a common trait, namely from “leaky barriers-to-neurotoxicity”. Furthermore, the AOP index is strongly predicted by the RR, indicating that increased zonulin coupled with lowered CMPAase and natural IgM protection may lead to increased leaky barriers-associated neurotoxicity. As explained previously, zonulin or pre-haptoglobin-2 (Hp2), a product of the Hp 2-2 gene, may loosen the tight junctions of the gut paracellular pathway leading to increased translocation of Gram-negative bacteria via the paracellular route [11,34,35]. This process is, subsequently, accompanied by increased activation of immune and oxidative (including the TRYCAT) pathways, for example via the Toll-Like Receptor (TLR)-4 complex [36]. There is evidence that LPS, TRYCATs, CCL11, and IL-6, may cause BBB permeabilization by disrupting the tight junctions of the paracellular BBB pathway [10,11]. Therefore, our results indicate that, in schizophrenia, peripheral immune activation via lowered RR and gut hyperpermeability is directly associated with BBB permeability and increased neurocognitive toxicity and that this may be a core axis in schizophrenia.

Even more important is that AOPs, the neurocognitive deficits, PHEM and negative symptoms, PMR and FTD are manifestations of a single trait, indicating that loosening of the barriers and increased neurotoxicity are directly associated with the substrate of the impairments in cognitive tests and the symptoms as well. As discussed previously, this indicates that a multitude of neurotoxic processes have damaged brain circuits including the prefrontal cortex, prefronto-temporal, prefronto-parietal, prefronto-striato-thalamic, hippocampal, and amygdalal neural circuits, the pre-supplementary motor area, and the supplementary motor area [1,37].

It is also important to note that PLS models permit to examine mediation as well as group differences in the PLS pathway model. As such, we discovered that the effects of the RR features on the HR-QoL were in fact mediated by AOPs, cognitive disturbances, and the symptomatome. GMA and permutations did now show major differences in the PLS model between men and women, although there was a significant difference in the path from the AOP-cognitome to the phenomenome, which was more pronounced in women than in men.

Finally, we have also explained how to disclose new classifications of schizophrenia by translating all feature sets (including RR, AOPs, and the phenome components) into latent variable scores and conducting unsupervised learning. Doing so, we successfully classified the patients into two classes whereby the first cluster (cluster 2) was characterized by lowered resilience (lowered IgM to OSEs and PON1 enzymatic activity) as well as significant higher AOP and phenome feature scores as compared with cluster 1, while both patient clusters were differentiated from controls by changes in IgM to zonulin, AOPs, cognitome, symptomatome, and phenomenome scores. Our clustering technique provided a class (cluster 2), which broadly agreed with deficit schizophrenia although our model was more restrictive than the SDS diagnosis. Neural networks showed that using the RR, AOP, and phenome feature scores as input variables allowed to predict membership to both clusters and controls with a 90.9% accuracy, thereby confirming that both schizophrenia subclasses are different nosological entities. Nevertheless, future research should include a larger number of patients to conduct predictive modeling and delineate the accuracy of optimized versions of our cluster classification.

The inductive reasoning and data-driven, nomothetic model built in the current study suggests that previous names given to the illness are not adequate. First, we showed that the DSM (and related ICD) diagnosis of schizophrenia comprises two qualitatively distinct classes and, therefore, it is not accurate to use one label such as schizophrenia to describe two different classes. Second, the dichotomy of “type 1” (positive) and “type 2” (negative) schizophrenia [38] is not adequate because our cluster 2 is associated with increased scores on all symptom features (PHEM, negative, PMR, FTD), AOPs, and cognitive disorders, indicating that these feature sets are mere manifestations of the same single trait, namely “the illness” and “overall severity of illness”. As discussed before, when there are increases in OSOS, all symptoms, but especially negative symptoms and PMR, become more prominent and shape a distinct clinical phenotype [2].

Moreover, the names given to delineate the illness are also not very adequate. First, Pick’s label “dementia praecox” is not useful because the phenome of cluster 2 contains many more features than a deficit in neurocognitive functions, and the latter does not even point towards dementia. Second, Bleuler’s label “schizophrenia” [39] is not accurate as it points towards a splitting of the mind-brain, while in fact there is no splitting, but aberrations in RR, AOPs, and the phenome. Third, the label “deficit schizophrenia” is also not adequate because cluster 2 is shaped by all features of the disorder and not merely negative symptoms as conceptualized by the SDS criteria [15]. The nomothetic network constructed here provides a new taxonomy of schizophrenia and, therefore, may be used to rename the illness [9]. This new name should stress the associations between increased neurotoxicity, cognitive impairments, and the symptomatome, and in addition should make a distinction between both cluster-derived classes. Thus, the clusters built herein could, for example, be described as Pervasive Psychosis due to NeuroCognitive Toxicity (PP-NCT) for cluster 2 and Simple Psychosis due to NCT (SP-NCT) for cluster 1.

The results of the present study should be discussed with respect to its limitations. First, this is a case control study and, therefore, one must be careful with causal interpretations. Nevertheless, the paths from the causome/protectome to the phenome (including AOPs, cognitome, and symptomatome) can be validated because it comprises genes and gene products (including CMPAase and zonulin) as well as deficits in natural IgM, which predispose to the AOPs, which are known to cause cognitive deficits and behavioral responses (see Introduction). Second, our results were obtained in stabilized patients and, therefore, cannot be extrapolated to patients in the acute phase of illness. A study is underway to create nomothetic models of the acute phase of psychosis. Third, future research should add magnetic resonance along with structural, functional, and spectroscopic assessments of the brainome to enrich our nomothetic model and to examine which brainome features belong to the AOP-cognitome-symptomatome phenotype [27]. Fourth, although we included 32 biomarkers and 16 clinical indicators in our model, larger samples with a wider array of genome, epigenome, and metabolome data as well as environmental and lifestyle factors should be added to build a more final model that should be cross-validated in larger, independent samples. We now employ our nomothetic model as a template to create larger mechanistic models that identify gene patterns and molecular signatures using curated databases coupled with pathway/network data analysis.

In conclusion, here we explained how to build a bottom-up, nomothetic model that integrates the features of psychosis (from cause-to-phenomenology) indicating that neurocognitive toxicity is a key component of the illness. This reification of a clinical diagnosis, the construction of new causal models containing all features, which are reduced to a fewer number of targeted feature sets, and the addition of curated research data projected into this new model, are awaited achievements that may radically change the way mental illnesses including psychosis are conceived.

## Figures and Tables

**Figure 1 brainsci-10-00645-f001:**
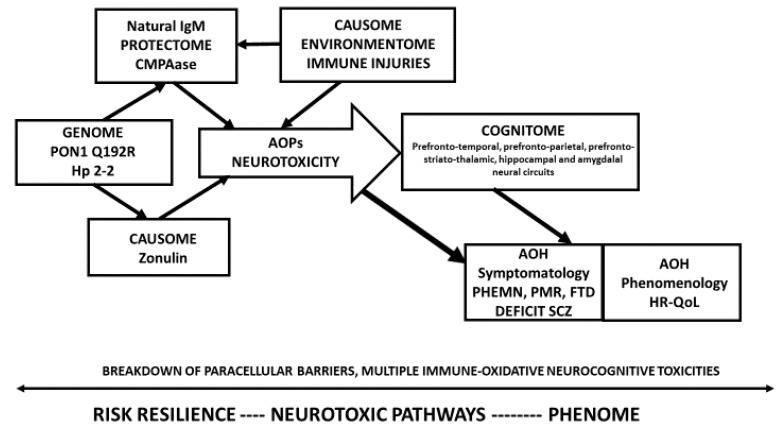
The causal reasoning model to be tested. PON1 Q192R: paraoxonase 1 (PON1) genotypes combined with PON1 4 (chloromethyl) phenol acetate (CMPA) ase activity; Hp 2-2: haptoglobin 2-2 genotype; AOP: adverse outcome pathways; AOH: adverse health outcomes; PHEMN: psychosis, hostility, excitation, mannerism, negative symptoms; SCZ: schizophrenia; HR-Qol: health-related quality of life.

**Figure 2 brainsci-10-00645-f002:**
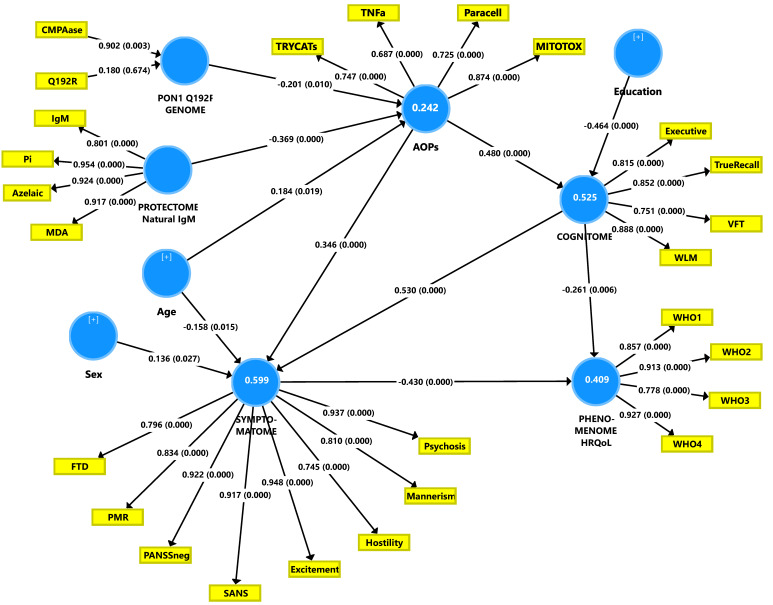
The first Partial Least Squares (PLS) path model constructed in this study.PON1 Q192R: paraoxonase 1 (PON1) genotypes combined with PON1 4(chloromethyl)phenyl acetate (CMPA)ase activity.MDA: malondialdehyde; Pi: phosphatidylinositol; AOPs: adverse outcome pathways; MITOTOX: index of multiple immune and oxidative toxicity; TNF: tumor necrosis factor; Paracell: index of paracellular route breakdown; TRYCATs: IgA to tryptophan catabolites; SANS: the Scale for the Assessment of Negative Symptoms; PANSSneg: the negative subscale of the Positive and Negative Syndrome Scale; FTD: formal thought disorders; PMR: psychomotor retardation. HRQoL: health-related quality of life; WHO 1-4: World Health Organization Quality of Life Instrument domains (domains 1-4). The white figures in blue circles indicate the explained variance.

**Figure 3 brainsci-10-00645-f003:**
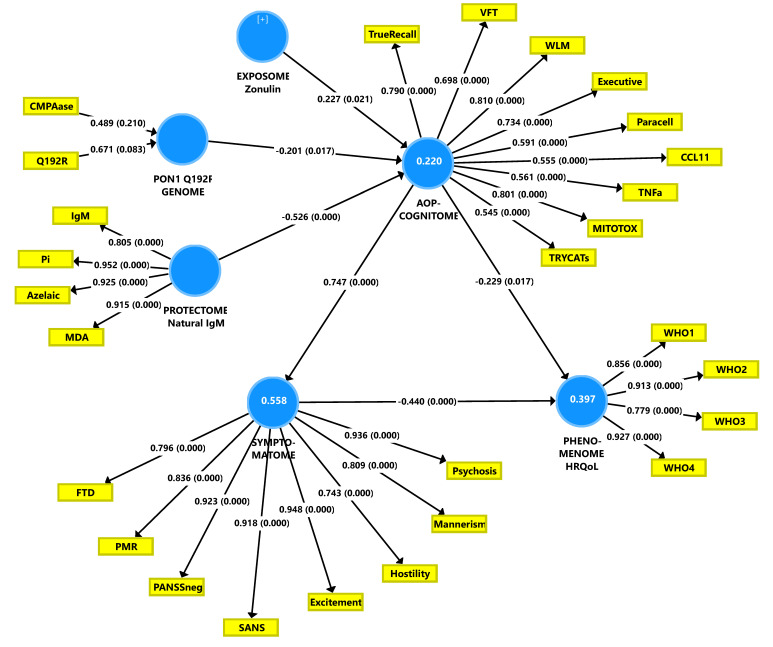
The second Partial Least Squares (PLS) path model constructed in this study. PON1 Q192R: paraoxonase 1 (PON1) genotypes with PON1 4 (chloromethyl) phenyl acetate (CMPA) ase activity. MDA: malondialdehyde; Pi: phosphatidylinositol. AOPs: adverse outcome pathways; MITOTOX: index of multiple immune and oxidative toxicities; TNF: tumor necrosis factor; Para cell: index of paracellular route breakdown; TRYCATs: IgA to tryptophan catabolites; WLM: Word List Memory; VFT: Verbal Fluency Test; SANS: the Scale for the Assessment of Negative Symptoms; PANSSneg: the negative subscale of the Positive and Negative Syndrome Scale; FTD: formal thought disorders; PMR: psychomotor retardation. HRQoL: health-related quality of life; WHO: World Health Organization Quality of Life Instrument. The white figures in blue circles indicate the explained variance.

**Figure 4 brainsci-10-00645-f004:**
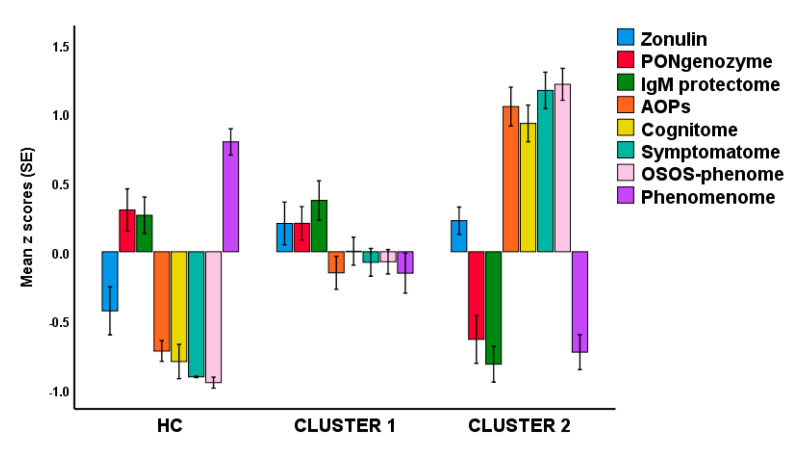
Latent variable scores in healthy controls (HC) and schizophrenia patients divided into two clusters. The scores reflect IgM to zonulin, a combination of CMPAase activity and the Q192R PON1 genotype (PONgenozyme), natural IgM to oxidative specific epitopes; AOPs: advanced outcome pathways; OSOS-phenome: overall severity of schizophrenia score which combines the AOP + cognitome + symptomatome feature scores.

**Figure 5 brainsci-10-00645-f005:**
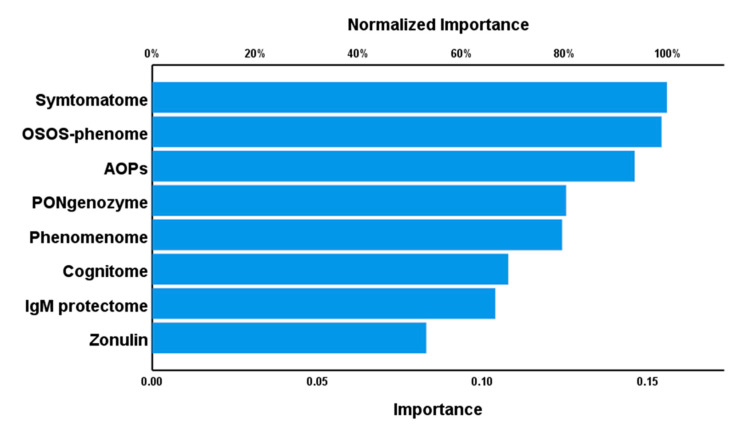
Results of neural networks showing the importance chart. Shown are the (relative) importance of the latent variable scores reflecting IgM to zonulin, a combination of CMPAase activity and the Q192R PON1 genotype (PONgenozyme), natural IgM to oxidative specific epitopes (IgM protectome); AOPs: advanced outcome pathways; OSOS-phenome: overall severity of schizophrenia score which combines the AOP + cognitome + symptomatome scores.

**Table 1 brainsci-10-00645-t001:** Socio-demographic data of the participants in this study divided according to the risk resilience (RR) index.

Variables	Normal RR ^A^ *n* = 29	Lowered RR ^B^ *n* = 57	Very Low RR ^C^ *n* = 34	F/χ^2^	df	*p*
Age (years)	37.0 (11.6)	39.5 (11.7)	42.9 (11.4)	F = 2.10	2/117	0.127
Sex (F/M)	18/11	25/22	14/20	χ^2^ = 4.14	2	0.126
BMI (kg/m2)	24.2 (4.2)	24.0 (4.7)	25.0 (4.9)	F = 0.37	2/114	0.690
Education (years)	14.3 (4.1) ^C^	13.3 (4.3) ^C^	11.2 (4.6) ^A,B^	F = 4.27	2/117	0.017
HC/NONDEF/DEFSCZ	15/11/3 ^C^	22/20/15 ^C^	3/9/22 ^A,B^	χ^2^ = 25.74	4	<0.001
TUD (No/Yes)	27/2	52/5	34/0	FET = 3.09	-	0.237
Employed (No/Yes)	9/20 ^C^	20/37 ^C^	21/13 ^A,B^	χ^2^ = 8.01	2	0.018
Q192R QQ/QR/RR	0/5/24 ^B,C^	1/28/28 ^A,C^	15/15/4 ^A,B^	FET = 52.15	-	<0.001
CMPAase (U/mL)	48.7 (9.2) ^B,C^	40.4 (8.1) ^A,C^	26.9 (10.3) ^A,B^	F = 48.17	2/117	<0.001
IgM sum OSEs (z score)	0.778 (0.783) ^B,C^	−0.059 (0.942) ^A,C^	−0.570 (0.743) ^A,B^	F = 19.32	2/117	<0.001
Total IgM (mg/dL)	147.6 (66.5) ^B,C^	101.0 (52.9) ^A^	87.8 (45.1) ^A^	F = 10.24	2/115	<0.001
IgM zonulin (z score)	−0.549 (1.053) ^B,C^	0.010 (0.990) ^A^	0.304 (0.791) ^A^	F = 6.82	2/117	0.002

Results are shown as mean (± SD); ^A,B,C^ post-hoc differences between the three categories. All results of analysis of variance (F), analysis of contingency tables (χ^2^), or Fisher’s exact test (FET). BMI: body mass index. HC/NONDEF/DEFSCZ: number of healthy controls (HC) and patients with (DEFSCZ) and without (NONDEF) deficit schizophrenia. TUD: tobacco use disorder. Q192R QQ/QR/RR: paraoxonase 1 (PON1) genotypes. CMPAase: PON1 CMPAase 4-(chloromethyl phenyl acetate-ase activity. OSEs: oxidative specific epitopes.

**Table 2 brainsci-10-00645-t002:** Measurements of the adverse outcome pathways (AOPs) and the cognitome in the participants divided according to a risk resilience (RR) index.

Variables (All in Z Scores)	Normal RR ^A^ *n* = 29	Lowered RR ^B^ *n* = 57	Very Low RR ^C^ *n* = 34	F	df	*p*
MITOTOX	−0.403 (0.175) ^C^	−0.168 (0.122) ^C^	0.565 (0.182) ^A,B^	8.20	2/105	<0.001
TNF-α	−0.466 (0.186) ^B,C^	0.010 (0.129) ^A^	0.367 (0.193) ^A^	4.90	2/105	0.008
IgA TRYCATs	−0.363 (0.183) ^C^	−0.106 (0.127)	0.342 (0.190) ^A^	3.68	2/105	0.029
CCL11	−0.244 (0.182) ^C^	−0.134 (0.126) ^C^	0.526 (0.189) ^A,B^	5.33	2/105	0.006
IgA PARA	−0.322 (0.176) ^C^	−0.138 (0.122) ^C^	0.485 (0.183) ^A,B^	5.74	2/105	0.004
Executive functions	0.288 (0.970) ^C^	0.067 (0.122) ^C^	−0.565 (0.182) ^A,B^	3.65	2/116	0.029
WLM	0.270 (0.069) ^C^	0.069 (0.126)	−0.433 (0.162) ^A^	4.75	2/116	0.010
WL Recall	0.190 (0.176) ^C^	0.080 (0.126) ^C^	−0.397 (0.162) ^A,B^	3.65	2/116	0.029
VFT	0.230 (0.179) ^C^	0.202 (0.129) ^C^	−0.508 (0.165) ^A,B^	6.63	2/116	0.002

Results are shown as mean (±SE) after covarying for age and sex (cognition) and age, sex, and body mass index (AOPs); ^A,B,C^ post-hoc differences between the three categories. All results of analysis of variance. MITOTOX: index of multiple immune and oxidative toxicities; TNF: tumor necrosis factor; IgA PARA: index of paracellular route breakdown. WLM: world list memory; VFT: verbal fluency test.

**Table 3 brainsci-10-00645-t003:** Measurements of the symptomatome and phenomenome in the participants divided according to a risk resilience (RR) index.

Variables (All in z Scores)	Normal RR ^A^ *n* = 29	Lowered RR ^B^ *n* = 57	Very Low RR ^C^ *n* = 34	F/χ^2^	df	*p*
SANS total	−0.435 (0.647) ^C^	−0.194 (0.792) ^C^	0.691 (1.205) ^A,B^	14.70	2/116	<0.001
PANSS negative	−0.528 (0.533) ^C^	−0.178 (0.831) ^C^	0.744 (1.146) ^A,B^	18.64	2/116	<0.001
Psychosis	−0.316 (0.897) ^C^	−0.121 (0.937) ^C^	0.470 (1.040) ^A,B^	6.11	2/116	0.003
Hostility	−0.294 (0.736) ^C^	−0.029 (1.082)	0.298 (0.992) ^A^	2.87	2/116	0.061
Excitement	−0.409 (0.804) ^C^	−0.123 (0.924) ^C^	0.556 (1.065) ^A,B^	9.23	2/116	<0.001
Mannerism	−0.440 (1.049) ^C^	−0.203 (1.365) ^C^	0.709 (1.753) ^A,B^	5.59	2/116	0.005
FTD	−0.118 (1.200) ^C^	−0.175 (0.860) ^C^	0.389 (0.949) ^A,B^	3.80	2/116	0.025
PMR	−0.528 (0.470) ^C^	−0.212 (0.755) ^C^	0.799 (1.207) ^A,B^	21.92	2/116	<0.001
WHO Qol Domain 1	0.256 (1.296) ^C^	0.081 (0.820) ^C^	−0.350 (0.908) ^A,B^	3.34	2/115	0.039
WHO Qol Domain 2	0.104 (1.079)	0.104 (0.937)	−0.257 (1.013)	1.60	2/115	0.207
WHO Qol Domain 3	0.036 (1.052)	0.066 (1.008)	−0.138 (0.957)	0.46	2/155	0.630
WHO Qol Domain 4	0.171 (1.049)	0.101 (0.985)	−0.309 (0.938)	2.39	2/155	0.096

Results of the symptomatome are shown as mean (± SE) after covarying for age and sex, or as mean (SD); ^A,B,C^ post-hoc differences between the three categories. All results of analysis of variance. SANS: the Scale for the Assessment of Negative Symptoms; PANSS negative: the negative subscale of the Positive and Negative Syndrome Scale; FTD: formal thought disorders; PMR: psychomotor retardation. WHO QoL: World Health Organization Quality of Life Instrument-Abbreviated version.

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
