# Peer review of "How to Construct a Bottom-Up Nomothetic Network Model and Disclose Novel Nosological Classes by Integrating Risk Resilience and Adverse Outcome Pathways with the Phenome of Schizophrenia"

_brainsci, 2020, doi:10.3390/brainsci10090645_

Round 1

Reviewer 1 Report

This is an extremely important study, which provides both empirical and conceptual insight into evidence based classification of mental disorders. As an alteranative of both conventional ex consensus nosological systems (with diagnosis essentially based exclusively on contestable evidence from subjective interviews and tests) and RDoC which is biologically orientied but still driven by expert commitment to definition of the domains of inquiry, Michael Maes and associates bring forward an ingenious project, the nomothetic network approach. 

That approach is blinded against the subjective nature of psychiatric diagnosis by adding multi-level number of variables as bio-markers of disease and their careful integration into stable networks to connect phenomena (symptomatome), cognitome and molecular pathways. 

By adding in vivo neuroimaging to such complex explanatory structures, current neuropsychiatry may enter into fundamental revision of its classification systems.

Author Response

@ANSWER: This is addressed in the text as:

Third, future research should add magnetic resonance structural, functional, and spectroscopic assessments of the “brainome” to enrich our nomothetic model and to examine which “brainome” features belong to the AOP-cognitome-symptomatome phenotype [27].

Reviewer 2 Report

The aim of the paper was to explain how to construct a reliable and replicable nomothetic network of schizophrenia.

Please specify more information about Figure 1. You said it's a pre-specified theoretical model. What are the references?

All the used tools are very well detailed in Methods. The phenomenome (HR QoL domains 1, 2, 3, and 4) was the final target which was predicted by all the other indicators.

You have split the population into three groups based on a visual binning method. Please specify what are the limits for very low RR, lower (or lowered ?!?) RR, and normal RR.

What are the limits of your research, if any?

Author Response

The aim of the paper was to explain how to construct a reliable and replicable nomothetic network of schizophrenia.

Please specify more information about Figure 1. You said it's a pre-specified theoretical model. What are the references?

@ANSWER: References are given. Addressed in the text as:

Hence, this study aims to explain how a) to construct a reliable and replicable nomothetic network of schizophrenia based on a theoretical model that was pre-specified based on our previous knowledge [1-5,9-14] (Figure 1)

All the used tools are very well detailed in Methods. The phenomenome (HR QoL domains 1, 2, 3, and 4) was the final target which was predicted by all the other indicators.

You have split the population into three groups based on a visual binning method. Please specify what are the limits for very low RR, lower (or lowered ?!?) RR, and normal RR.

@ANSWER: The thresholds as defined through visual binning are now defined in the text:

we have split the population into three groups based on a visual binning method using z=-0.53 and z=0.80 as cut-off-values.

What are the limits of your research, if any?

@ANSWER: Limitations are discussed:

 Nevertheless, future research should include a larger number of patients to conduct predictive modeling and delineate the accuracy of optimized versions of our cluster classification.

And:

The results of the present study should be discussed with respect to its limitations. First, this is a case control study and, therefore, one must be careful with causal interpretations. Nevertheless, the paths from the causome/protectome to the phenome (including AOPs, cognitome and symptomatome) can be validated because it comprises genes and gene products (including CMPAase and zonulin) as well as deficits in natural IgM which predispose to the AOPs, which are known to cause cognitive deficits and behavioral responses (see Introduction). Second, our results were obtained in stabilized patients and, therefore, cannot be extrapolated to patients in the acute phase of illness. A study is underway to create nomothetic models of the acute phase of psychosis. Third, future research should add magnetic resonance structural, functional, and spectroscopic assessments of the brainome to enrich our nomothetic model and to examine which brainome features belong to the AOP-cognitome-symptomatome phenotype [27]. Fourth, although we included 32 biomarkers and 16 clinical indicators in our model, larger samples with a wider array of genome, epigenome, and metabolome data as well as environmental and lifestyle factors should be added to build a more final model that should be cross-validated in larger, independent samples.